# Detection and Evaluation for High-Quality Cardiopulmonary Resuscitation Based on a Three-Dimensional Motion Capture System: A Feasibility Study

**DOI:** 10.3390/s24072154

**Published:** 2024-03-27

**Authors:** Xingyi Tang, Yan Wang, Haoming Ma, Aoqi Wang, You Zhou, Sijia Li, Runyuan Pei, Hongzhen Cui, Yunfeng Peng, Meihua Piao

**Affiliations:** 1School of Nursing, Chinese Academy of Medical Sciences & Peking Union Medical College, Beijing 100144, China; tangxingyi@pumc.edu.cn (X.T.); wangyan@nursing.pumc.edu.cn (Y.W.); haoming_ma@student.pumc.edu.cn (H.M.); aoqi_wang2023@163.com (A.W.); pumc_youzhou@163.com (Y.Z.); sijia_li1101@163.com (S.L.); phenitixpumc@163.com (R.P.); 2School of Computer and Communication Engineering, University of Science and Technology Beijing, Beijing 100083, China; chz_367@163.com (H.C.); pengyf@ustb.edu.cn (Y.P.)

**Keywords:** cardiopulmonary resuscitation, motion capture, three-dimensional, high-quality, arm posture, training

## Abstract

High-quality cardiopulmonary resuscitation (CPR) and training are important for successful revival during out-of-hospital cardiac arrest (OHCA). However, existing training faces challenges in quantifying each aspect. This study aimed to explore the possibility of using a three-dimensional motion capture system to accurately and effectively assess CPR operations, particularly about the non-quantified arm postures, and analyze the relationship among them to guide students to improve their performance. We used a motion capture system (Mars series, Nokov, China) to collect compression data about five cycles, recording dynamic data of each marker point in three-dimensional space following time and calculating depth and arm angles. Most unstably deviated to some extent from the standard, especially for the untrained students. Five data sets for each parameter per individual all revealed statistically significant differences (*p* < 0.05). The correlation between Angle 1′ and Angle 2′ for trained (r_s_ = 0.203, *p* < 0.05) and untrained students (r_s_ = −0.581, *p* < 0.01) showed a difference. Their performance still needed improvement. When conducting assessments, we should focus on not only the overall performance but also each compression. This study provides a new perspective for quantifying compression parameters, and future efforts should continue to incorporate new parameters and analyze the relationship among them.

## 1. Introduction

Out-of-hospital cardiac arrest (OHCA) remains a significant public health challenge, with the effectiveness of cardiopulmonary resuscitation (CPR) being a critical determinant in improving patient survival rates [1,2]. In the majority of OHCA cases, bystanders were present at the scene. Research has demonstrated a statistically significant improvement in survival rates for individuals who received CPR from these bystanders compared to those who did not receive such intervention [3]. Despite advancements in public awareness and attitudes towards CPR in recent years, the overall survival rates post-OHCA remain suboptimal [3,4]. The effectiveness of CPR, particularly when performed with high proficiency, is vital for the successful resuscitation of patients [5]. Furthermore, the efficacy of resuscitation is notably influenced by the educational quality of CPR training [6], underscoring the necessity of enhancing public access to high-quality CPR training programs, as advocated in various international guidelines [7,8].

CPR training predominantly utilizes training simulators, such as the Resusci-Anne manikin, to facilitate skill acquisition. These simulators are instrumental in measuring specific parameters, including compression rate, depth, and ventilation volume [9], while the instructors provide direct observation and assessment of techniques not captured by the manikin, notably the positioning and movement of the arms [10]. For instance, fully extended arms during CPR allow for more effective use of the rescuer’s body weight, facilitating the achievement of the necessary compression depth while conserving energy. Consequently, the bending of the arms can adversely impact CPR quality by influencing key factors such as compression depth, frequency, and the rescuer’s endurance, potentially leading to early fatigue. Additionally, a frequent challenge in CPR training is the limited instructors’ abilities to evaluate multiple trainees concurrently. This scenario often leads to discrepancies in training quality, as the feedback provided by instructors can vary significantly based on their individual expertise and experience.

Multimodal Learning Analytics (MMLA) present innovative solutions to these limitations by providing automated, real-time feedback [11]. This technology has traditionally been limited to computer-based interfaces but is increasingly applicable to practical skill acquisition in physical spaces, facilitated by advancements in sensor technology and mobile computing [12]. These developments allow for the capture of a wide range of human activities and interactions necessary for psychomotor learning, offering more detailed and comprehensive data for analysis.

In order to enhance the efficiency and the accuracy of CPR training, an increasing number of studies have begun to utilize devices that provide automated and objective CPR feedback, including pressure sensors [13], accelerometers [14,15], video [16], and motion capture devices (like Kinect) [17] to assist instructors in evaluating the CPR process more effectively. Most of these studies focus on key parameters like compression depth, rate, and proper release [17,18]. However, challenges persist, with some devices potentially disrupting the training process or lacking in precision, particularly in the quantification of arm postures, a crucial aspect of effective CPR technique.

In this research, we explored the integration of MMLA into CPR training to improve instructional quality and efficiency. We utilized three-dimensional (3D) motion capture technology for objective assessment of key CPR techniques including each compression depth, and arm posture parameters typically subject to human instructor evaluation. We also investigated the relationship between depth and arm angles to guide students in enhancing CPR quality.

## 2. Materials and Methods

To accurately monitor and assess the quality of each compression, we utilized motion capture devices to capture 3D dynamic data during CPR operations. We calculated the depth and arm angles, exploring the relationship between them.

### 2.1. Participants

In order to analyze various qualities of CPR, we selected four undergraduate students, with Students 1, 3, and 4 having undergone at least three CPR training courses, while Student 2 has not received any. All signed an informed consent letter, which included details of the experiment, data privacy protection, the freedom to withdraw, and other relevant information.

### 2.2. Experimental Setup

We used the Resusci-Anne^®^ manikin to simulate an adult cardiac arrest and the infrared light spot motion capture system (Mars series, Nokov, Beijing, China) to obtain the marker point motion data at a collection frequency of 60 Hz. This system includes infrared light cameras and data visualization software (Motion Kinematics & Kinetics Analyzer, Mokka, 0.6.0.0). The motion capture device can record the dynamic coordinates (millimeters) of each marker point in the X, Y, and Z axes in three-dimensional space over time (seconds) by infrared light cameras arranged around the room. Based on expert consultations, a total of 12 marker points were applied to the top of the head, anterior head point, posterior head point, left/right acromion, left/right lateral epicondyle of the humerus, midpoint of the line connecting the left/right ulnar styloid process with the radial styloid process, and spines (Figure 1a). The data visualization software illustrates the reproduction of the marker points by importing the coordinate data of the X, Y, and Z axes into it (Figure 1b). Among them, data from four marker points—left acromion (marker point A), left lateral epicondyle of the humerus (marker point B), midpoint of the line connecting the left ulnar styloid process with the radial styloid process (marker point C), and right acromion (marker point D)—were used for subsequent data analysis, while the remaining marker points were used as reference points for assessing body posture during review. We also used a videotape recorder to record the entire process.

### 2.3. Procedures

We used the Resusci-Anne^®^ manikin to simulate an adult cardiac arrest but did not retrieve any information from it. Each participant performed 5 cycles of compression on it, and each cycle included approximately 30 compressions. During the process of compression, the participants’ hands were asked not to leave the chest of the manikin. We recorded each participant separately. To protect the privacy of participants, recorded videos were only used for initial data correction and would eventually be removed from the dataset. According to the motion capture device, we collected data from four instances of the CPR procedure. The dataset comprised the coordinates (millimeters) of various marker points changing over time (seconds) in the X, Y, and Z axes. For each participant, we calculated compression depth, the angle between the left upper arm and lower arm at the elbow (referred to as Angle 1) (Figure 2a), and the angle between both elbows and the ground (referred to as Angle 2) (Figure 2b).

### 2.4. Data Processing

For each motion capture dataset, according to the reproduction of the marker points by data visualization software, we manually removed interrupted compression data, extracted five sets of cycles, including time and the coordinates of four marker points—left acromion (marker point A), left lateral epicondyle of the humerus (marker point B), midpoint of the line connecting the left ulnar styloid process with the radial styloid process (marker point C), and right acromion (marker point D)—on the X, Y, and Z axes. When encountering missing points or other discrepancies, we manually corrected the data according to the recorded video and the reproduction in data visualization software from that time.

Subsequently, we used the Z-axis coordinate data of marker point C as a reference. This formed a dataset showing a ‘decrease–increase’ pattern over time (during compression), where each ‘decrease–increase’ set represents one compression. This corresponded to the extraction of ‘peak–lowest-peak’ data for each set. Within each ‘peak–lowest-peak’ data trio, the difference between the first peak data and the subsequent nearest low data represented the depth of one compression measured in centimeters. According to the three-dimensional coordinates, we calculated Angle 1 and 2 (each peak or lowest point) using vector Equations (1) and (2), respectively.
(1)$$\theta 1=\mathrm{arcos}\left[\frac{\;\left(\mathrm{XA}-\mathrm{XB},\;\mathrm{YA}-\mathrm{YB},\;\mathrm{ZA}-\mathrm{ZB}\right)\;\cdot \;\left(\mathrm{XC}-\mathrm{XB},\;\mathrm{YC}-\mathrm{YB},\;\mathrm{ZC}-\mathrm{ZB}\right)}{\sqrt{{\left(\mathrm{XA}-\mathrm{XB}\right)}^{2}+{\left(\mathrm{YA}-\mathrm{YB}\right)}^{2}+{\left(\mathrm{ZA}-\mathrm{ZB}\right)}^{2}\;}\sqrt{{\left(\mathrm{XC}-\mathrm{XB}\right)}^{2}+{\left(\mathrm{YC}-\mathrm{YB}\right)}^{2}+{\left(\mathrm{ZC}-\mathrm{ZB}\right)}^{2}}}\right]$$
(2)$$\theta 2=\mathrm{arcos}\left\{\frac{\left(\mathrm{XC}-\mathrm{XA}\right)\left(\mathrm{YD}-\mathrm{YA}\right)-\left(\mathrm{YC}-\mathrm{YA}\right)\left(\mathrm{XD}-\mathrm{XA}\right)}{\sqrt{{\left[\left(\mathrm{YC}-\mathrm{YA}\right)\left(\mathrm{ZD}-\mathrm{ZA}\right)-\left(\mathrm{ZC}-\mathrm{ZA}\right)\left(\mathrm{YD}-\mathrm{YA}\right)\right]}^{2}+{\left[\left(\mathrm{ZC}-\mathrm{ZA}\right)\left(\mathrm{XD}-\mathrm{XA}\right)-\left(\mathrm{XC}-\mathrm{XA}\right)\left(\mathrm{ZD}-\mathrm{ZA}\right)\right]}^{2}+{\left[\left(\mathrm{XC}-\mathrm{XA}\right)\left(\mathrm{YD}-\mathrm{YA}\right)-\left(\mathrm{YC}-\mathrm{YA}\right)\left(\mathrm{XD}-\mathrm{XA}\right)\right]}^{2}}}\right\}$$

### 2.5. Data Analysis

For each parameter calculated above, depth includes the depth of each compression, while angle includes the angle of the lowest and highest points of each compression; we drew line graphs or scatter plots encompassing ‘Depth (cm)–Time (s)’, ‘Angle 1 (°)–Time (s)’, and ‘Angle 2 (°)–Time (s)’. Using SPSS 26.0 for statistical analysis, median (P25, P75) about depth, Angle 1′ (=|180—Angle 1|), and 2′ (=|90—Angle 2|) for each student were calculated. The differences among five data sets for each parameter per individual were analyzed using the Kruskal–Wallis H test. The correlation among depth, Angle 1′, and Angle 2′ was analyzed by Spearman’s rank correlations. The values of *p* < 0.05 were considered statistically significant.

## 3. Results

The results respectively display parts of scatter plots or line graphs of ‘Depth (cm)–Time (s) (Figure 3)’, ‘Angle 1 (°)–Time (s) (Figure 4)’, and ‘Angle 2 (°)–Time (s) (Figure 5)’, with the red line representing the standard reference values. According to CPR standards [7,19,20], compression depth should be 5–6 cm, and the arms should be straight and perpendicular to the ground. A measurement of Angle 1 with markers A, B, and C equal to 180° indicates that the elbows were locked and the arms were kept straight. Angle 2, between the plane formed by markers A, C, and D, and the ground being perpendicular, suggests that the shoulders were directly over the sternum.

Student 2 had an unstable compression depth, with four sets consistently below the standard and one set exceeding it. Students 1, 3, and 4 exhibit relatively stable compression depth, though generally higher than the standard (Figure 3). Angle 1 of Student 2 is unstable, particularly in cycles 1, 2, and 5, deviating significantly from 180°. Conversely, Student 1 maintains more stability with fewer deviations from 180°. Although Students 3 and 4 display stability, they consistently deviate from 180° (Figure 4). Angle 2 of Student 2 is unstable, especially in cycles 3 and 4, showing frequent deviations from 90°. In contrast, Students 1 and 3 maintain relatively stable with fewer deviations from 90°. Despite stability, Student 4 consistently deviates from 90°, with a tendency for both arms to lean forward at the beginning of each cycle (Figure 5).

The median (P25, P75) of depth, Angle 1′, and Angle 2′ for Student 1 is 7.06(6.85, 7.27), 11.53(9.88, 13.17), and 8.98(7.50, 10.24), respectively. For other students, the detailed information is shown in Table 1. All data did not meet normal distribution, so we employed the Kruskal–Wallis H test. Five data sets for each parameter per individual all revealed statistically significant differences (*p <* 0.05) (Table 1).

Student 1 received training and performed relatively consistently, while Student 2 did not receive training and was less stable. Therefore, we selected Student 1 and Student 2 for the analysis of the correlation among depth, Angle 1′, and Angle 2′. For Student 1, there was a significant positive correlation between Angle 1′ and Angle 2′ (r_s_ = 0.203, *p* < 0.05) (Table 2). For Student 2, there was a significant negative correlation between Angle 1′ and Angle 2′ (r_s_ = −0.581, *p* < 0.01). In addition, Angle 1′ (r_s_ = −0.209, *p* < 0.01) and Angle 2′ (r_s_ = 0.467, *p* < 0.01) were significantly correlated with depth, respectively (Table 3).

## 4. Discussion

The main purpose of this study is to explore the possibility of utilizing the motion capture device to accurately assess CPR quality parameters, including not only depth but also the rarely quantified arm postures. Additionally, we investigate the relationship between depth and arm angles to guide students in enhancing CPR training quality.

This study found that untrained Student 2 exhibited instability and relatively poor performance, struggling to maintain both straight arms and arms perpendicular to the ground simultaneously. Trained Students 1, 3, and 4 showed relative stability but still required correction in compression depth and arm postures. For example, their compression depth tended to be higher on average. Students may exhibit unconscious patterns, such as the consistent forward tilt observed in Student 4’s Angle 2 at the beginning of each cycle. So, it is necessary to correct students’ habitual errors through standardized evaluation and training. Furthermore, there were significant differences in the data for each parameter across the five cycles for each student. This study provides a preliminary exploration into the effectiveness of motion capture systems in accurately monitoring and assessing various CPR quality parameters. In the future, the systems could be used to provide targeted feedback to students, enhancing their awareness and performance.

More studies use Kinect or attach a sternum sensor integrated with accelerometers or pressure sensors to the compression point for monitoring compression depth and rate. However, if we move the Kinect to a different location, we will achieve different values of the sensors [10]. Therefore, the rescuer needs to stand in a specific position. Thick sensors may shift during compressions, potentially causing skin injuries for the rescuer or the patient [13]. Additionally, using springs as pressure sensors requires greater compression force, which can lead to fatigue. However, the motion capture device can effectively avoid the aforementioned issues because it is equipped with cameras arranged around the room, providing not only a larger field of view but also not affecting the rescuer’s experience. Certainly, we should continue to explore the accuracy of the device and compare it with other devices.

A patient with cardiac arrest may require uninterrupted and effective CPR for a duration of at least 30–45 min [21]. It is a maneuver that demands a lot of physical stamina, while fatigue may affect the quality of chest compression [22,23]. For Student 2, there was a significant negative correlation between Angle 1′ and depth, which meant that more bent arms tended to have lower compression depth. Straight arm postures are beneficial to avoid early fatigue [24] and improve the quality [25]. However, there was a significant positive correlation between Angle 2′ and depth, which contradicted our hypothesis. In theory, if the direction of the pressing force deviates from the vertical direction, the compression force will be dispersed, resulting in a shallower compression depth [20]. Correct arm posture may help rescuers effectively leverage the gravity to generate the necessary pressure and reduce damage to other sternum structures. Therefore, we should pay attention to the arm posture during CPR training to help reduce muscle fatigue. We only studied five consecutive sets of compressions, lasting approximately two minutes, which is insufficient to observe changes in compression parameters and quality over time. Future research could extend the compression duration to explore the effects of time.

Regarding the arm posture, some studies quantified the straightening of the arm using 2D data through video [24] or electromyogram [10]. Although they can monitor arm posture, they still have some limitations. For instance, 2D videos may encounter difficulties in accurate identification due to the occlusion of individuals. Additionally, electromyography cannot precisely quantify arm angles and is unable to monitor the angle between the arm and the ground. The current approach mainly relies on instructors’ visual assessments [26]. They often provide a holistic evaluation by checklist, generalizing the overall compression postures throughout the entire process without specifying each compression [27,28]. Through the comparison of five sets of data within individuals, we found significant statistical differences, indicating that each performance varied over time during CPR operations for each person. In particular, considerable variation was observed across the different cycles for Student 2. Therefore, relying on instructors may cause a lack of accuracy for each compression and visual error. Our study provides a new perspective for quantifying arm angles using 3D data with a larger field of view, facilitating instructors to more accurately assess each compression’s posture while adding the parameter of whether both arms are vertical to the ground. Compared to Kinect, our device fixes cameras around the room, allowing multiple cameras to simultaneously monitor more students. And it can offer comprehensive and personalized evaluations, guiding students in targeted improvements, while instructors can focus on other aspects.

In the analysis of Angle 1′ and 2′, a distinct positive correlation was identified in the case of Student 1, in contrast to a negative correlation observed for Student 2. This disparity could potentially be attributed to the prior CPR training of Student 1, enabling more effective coordination in maintaining both arm extension and the vertical orientation of the arms relative to the ground. He also exhibited consistent angular stability throughout the five cycles. However, the relationship between the depth and arm angles remained unclear. On the other hand, Student 2, lacking prior CPR training, exhibited considerable variability in angular measurements across the cycles and was unable to simultaneously achieve the desired arm extension and vertical alignment. As some studies found, untrained people performed poorly compared to those who had undergone training [29,30]. Student 2’s instability in compression may be due to lack of training and inability to self-assess whether the operation is correct, or it could be due to improper technique resulting in exertion. In this feasibility study, our sample size was insufficient. Although a correlation between compression depth and Angles 1′ and 2′ was observed, further investigation is needed to analyze this relationship in detail. Future studies should aim to improve the experimental approach, expand the participant types and sample sizes, and lengthen the duration of CPR practice to better replicate real-life scenarios. Such studies would enable a more comprehensive analysis of the individual and collective impacts of Angle 1 (arm extension) and Angle 2 (vertical alignment of arms) on compression quality. Additionally, besides focusing on the quality of compressions, it is essential to consider the sensations of the rescuers, such as fatigue, by utilizing electromyography to evaluate the exertion of the muscles in both arms.

Our observations indicated that the adherence of arm postures to the ideal angles of 180° or 90° was not consistent, which could be attributed to variations in anatomical structure or the positioning of markers [24]. The previous research also mentioned that it was impossible to maintain completely straight arms during compression [20]. Identifying a standard assessment range for these angles is a pertinent area for future research. Moreover, our experimental methodology presents opportunities for enhancement, including the reduction in the number of markers to prevent marker loss and the standardization of marker placement to decrease systemic inaccuracies. Additionally, incorporating measurements for other relevant parameters, such as ventilation, could enrich the study’s comprehensiveness. Many studies focus on the impact of real-time feedback on CPR quality, but there is still controversy [31,32]. Feedback mainly includes the presentation of compression depth and rate, neglecting feedback on arm angles. Motion capture devices offer potential for further research in this area. Looking forward, the application of motion capture technology in medical training has the potential to be expanded beyond CPR, employing sophisticated algorithms to lessen the instructional load and furnish students with immediate, personalized feedback and guidance.

## 5. Conclusions

In this experiment, we explored the possibility of using a motion capture system to collect three-dimensional data for assessing CPR performance parameters. Apart from depth, we introduced two angles, arm straightening, and both arms being vertical to the ground, effectively addressing the limitations of instructor visual assessments. At the same time, we conducted a preliminary analysis of the relationship among these parameters. Compared to untrained students, trained students demonstrated better performance in terms of stability and simultaneously considering the angles of both arms. This approach quantified each compression, assisting instructors in directing their attention to other aspects and aiding students in targeted improvements to enhance CPR quality. In the future, it is imperative to further refine our solutions to enhance the comprehensive and accurate application of motion capture devices in monitoring and assessing CPR or other medical procedures.

## Figures and Tables

**Figure 1 sensors-24-02154-f001:**
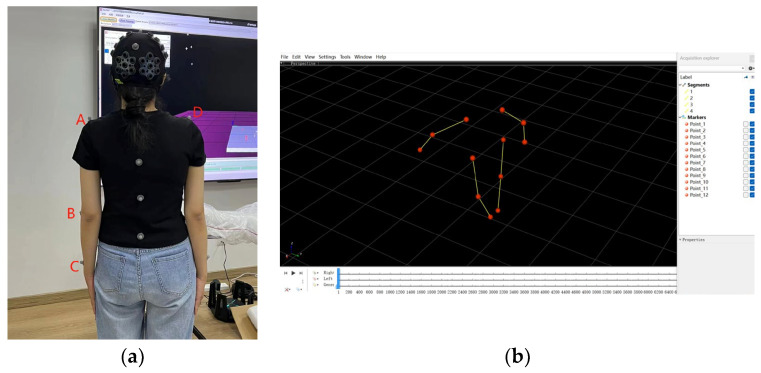
The locations of 12 markers. (**a**) Real locations in the body of the participant: left acromion (marker point A), left lateral epicondyle of the humerus (marker point B), midpoint of the line connecting the left ulnar styloid process with the radial styloid process (marker point C), and right acromion (marker point D); (**b**) their reproductions in the interface of data visualization software.

**Figure 2 sensors-24-02154-f002:**
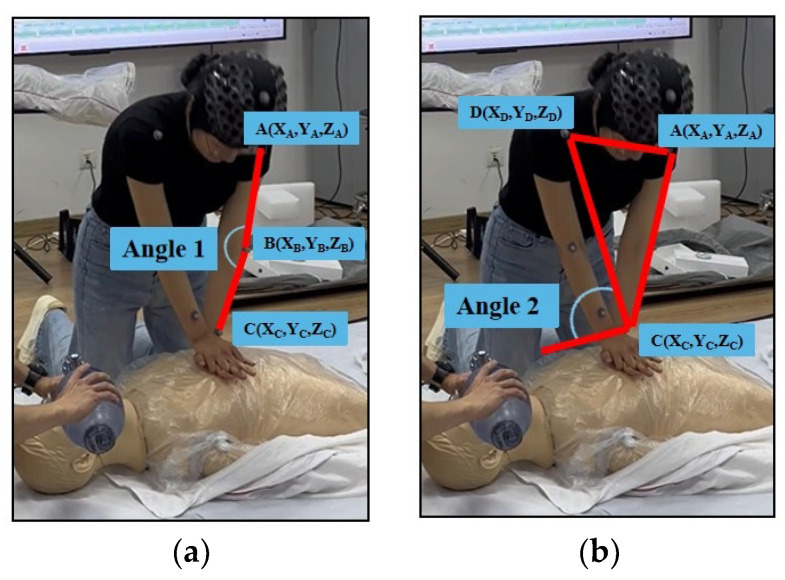
Angle 1 and Angle 2. (**a**) Marker points A, B, and C form Angle 1; (**b**) marker points A, C, and D form a plane, and the plane and the ground form Angle 2.

**Figure 3 sensors-24-02154-f003:**
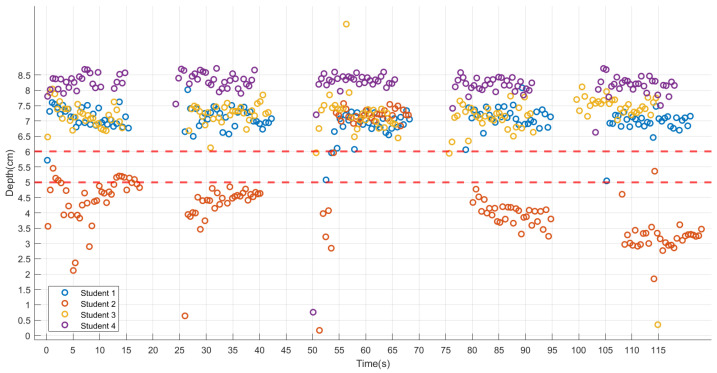
‘Depth (cm)–Time (s)’ for each student. The red lines represent the standard reference values (5–6 cm).

**Figure 4 sensors-24-02154-f004:**
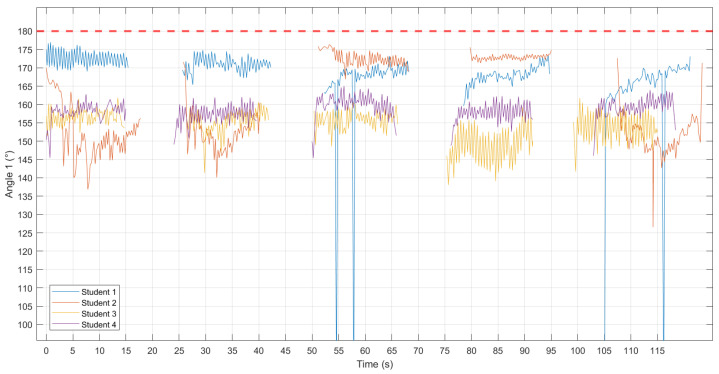
‘Angle 1 (°)–Time (s)’ for each student. The red line represents the standard reference values (180°).

**Figure 5 sensors-24-02154-f005:**
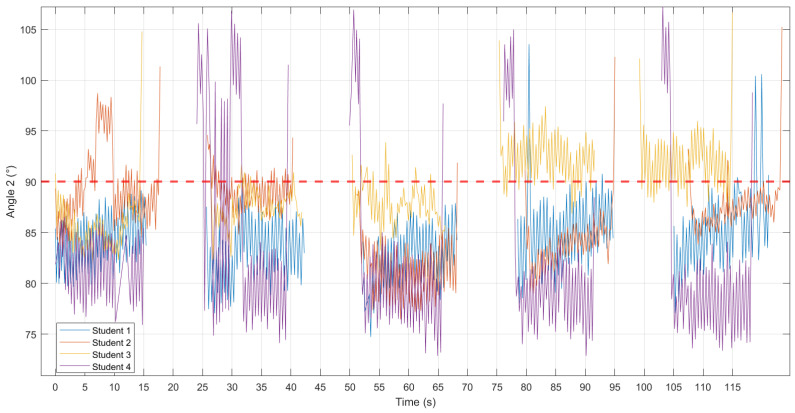
‘Angle 2 (°)–Time (s)’ for each student. The red line represents the standard reference values (90°).

**Table 1 sensors-24-02154-t001:** The median (P25, P75) and difference for depth, Angle 1′, and 2′.

Student	Cycle	Depth (cm)		Angle 1′ (°)		Angle 2′ (°)	
		Median (P25, P75)	H(*p*)	Median (P25, P75)	H(*p*)	Median (P25, P75)	H(*p*)
1	1	7.14(6.89, 7.43)	14.00 (0.007)	9.71(9.39, 10.06)	63.73 (<0.001)	8.84(7.45, 9.96)	20.29 (<0.001)
	2	7.10(6.89, 7.39)		10.32(9.84, 11.41)		9.57(8.65, 11.29)	
	3	6.89(6.70, 7.10)		12.35(11.96, 13.66)		9.84(8.48, 10.83)	
	4	7.13(6.94, 7.36)		12.91(11.35, 14.12)		8.12(6.90, 9.65)	
	5	7.01(6.83, 7.12)		13.15(10.89, 16.61)		8.07(6.96, 9.26)	
	Total	7.06(6.85, 7.27)		11.53(9.88, 13.17)		8.98(7.50, 10.24)	
2	1	4.69(4.15, 5.00)	87.08 (<0.001)	27.76(24.89, 30.88)	120.05 (<0.001)	4.10(2.74, 5.82)	108.60 (<0.001)
	2	4.48(4.12, 4.62)		28.80(25.47, 32.51)		2.63(2.06, 3.31)	
	3	7.18(6.92, 7.27)		9.14(8.13, 9.70)		11.63(10.60, 12.32)	
	4	4.05(3.72, 4.16)		7.80(7.37, 8.02)		7.23(6.40, 8.71)	
	5	3.23(2.97, 3.34)		30.56(27.14, 32.82)		3.64(2.95, 4.44)	
	Total	4.33(3.58, 4.98)		24.18(8.68, 30.16)		4.92(3.10, 8.48)	
3	1	7.11(6.91, 7.39)	22.25 (<0.001)	25.15(24.17, 25.99)	102.88 (<0.001)	6.26(4.48, 7.08)	96.108 (<0.001)
	2	7.30(7.10, 7.45)		26.43(24.94, 27.69)		3.44(2.32, 4.33)	
	3	7.22(6.87, 7.41)		26.09(24.63, 27.96)		3.57(2.67, 4.55)	
	4	7.15(6.80, 7.31)		36.07(33.80, 37.54)		0.84(0.33, 1.48)	
	5	7.54(7.29, 7.68)		29.99(28.47, 31.45)		1.09(0.55, 1.69)	
	Total	7.26(7.02, 7.47)		27.72(25.38, 31.51)		2.57(1.04, 4.42)	
4	1	8.27(8.08, 8.47)	11.58 (0.021)	22.31(22.16, 23.30)	70.53 (<0.001)	12.01(11.13, 12.51)	52.61 (<0.001)
	2	8.36(8.16, 8.56)		24.67(23.86, 25.36)		13.85(13.05, 14.96)	
	3	8.34(8.21, 8.43)		20.63(19.88, 22.76)		13.99(13.47, 14.91)	
	4	8.17(8.04, 8.34)		23.85(23.57, 24.98)		14.16(12.88, 15.04)	
	5	8.20(8.03, 8.30)		21.78(21.20, 23.14)		15.08(14.49, 15.76)	
	Total	8.27(8.09, 8.41)		23.21(21.70, 24.19)		14.02(12.76, 14.10)	

**Table 2 sensors-24-02154-t002:** The correlation among depth, Angle 1′, and Angle 2′ for Student 1.

	Angle 1′ (°)	Angle 2′ (°)
Depth (cm)	0.017	0.079
Angle 1′ (°)		0.203 *

* *p <* 0.05.

**Table 3 sensors-24-02154-t003:** The correlation among depth, Angle 1′, and Angle 2′ for Student 2.

	Angle 1′ (°)	Angle 2′ (°)
Depth (cm)	−0.209 **	0.467 **
Angle 1′ (°)		−0.581 **

** *p <* 0.01.

## Data Availability

The datasets used and/or analyzed during the present study are available from the corresponding author upon reasonable request.

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
