# Peer review of "Detection and Evaluation for High-Quality Cardiopulmonary Resuscitation Based on a Three-Dimensional Motion Capture System: A Feasibility Study"

_sensors, 2024, doi:10.3390/s24072154_

Round 1

Reviewer 1 Report

Comments and Suggestions for Authors

General comments:

The article is interesting, original and well-done. I recommend precising a few points before the article would be ready for publication in the journal.

The main point is to ensure the possibility for readers to reproduce the work. For that, the authors should give more pieces of information about the protocol, the software and the method of analysis. As it is now, I fear the reproducibility would be hard.

The other, minor points are:

-       My understanding of the project is a feasibility study. I think that the authors should state it like that.

-       It is not clear to me what feedback (if any) the authors retrieve from the manikin. I understand that no information was retrieved but this should be clarified because having precise efficacy feedback from the manikin would be of interest.

-       An interesting point for me is the stability of each student, that I think should be discussed in the discussion part.

-       When performing CPR we always fear that people may lose strength over time. Why not having measured the impact of time on the CPR? This should also be added in the discussion.

Specific comments:

Method part:

1.     Section 2.3: please add which variables were retrieved from the manikin (depth of compression, speed,…)

2.     Line 122: what do “Nokov” refer to? Is it the sensor system or the manikin.

The authors should try to remain consistent with the terms employed to describe the devices to ease the reading.

Results part:

3.     Line 152: the sentence “Student 2 had the unstable compression depth of with four sets…” should be corrected as the use of the expression “depth of with” does not seem appropriate.

4.     Line 153: “StudentS 1, 3, and 4”

5.     Lines 171-176: I am not sure that those lines are really useful since it is hard to read and produced more clearly in the table 1

6.     Line 200: “Trained StudentS”

Discussion part:

7.     See in the general comments my 2 remarks.

8.     I think that the possibilities offered by such a device should be more discussed, such as the possibility to correct oneself thanks to the presentation of the result, …

Reviewer 2 Report

Comments and Suggestions for Authors

The main purpose of this study is to explore the possibility of utilizing the motion capture device to accurately assess CPR quality parameters. The relationship between depth and arm angles to guide students in enhancing CPR training quality is investigated. On the otherhand, 

_According to CPR standards, it is clear that compression depth should be 5-6 cm, and the arms should be straight [7, 19]. In this point of view, the relationship between the depth and Angle1 is important. On the other hand, the relationship between Angle 2 and the depth is not clear.
_In the experimental measurement, untrained student is only one among four students. In this experimental setup, it is difficult to conclude general dfference between untrained students and trained students. Moreover, to have common result, the number of four subjects is too small.
_The relationship between the depth and arm angles remained unclear. Further investigation is needed to analyze a correlation between compression depth and angles 1’ and 2’ to understand why the arms should be straight.

Reviewer 3 Report

Comments and Suggestions for Authors

Dear Editor & Authors,

First of all, I think that the manuscript entitled: “Detection and Evaluation for High-Quality Cardiopulmonary Resuscitation Based on Three-Dimensional Motion Capture System: A Preliminary Test” submitted for publication in the Sensors Journal (MDPI) has both practical and scientific interest.

More specifically:

Ø  This test aimed to explore the possibility of using a three-dimensional motion capture system to accurately and effectively assess CPR operations, particularly about the non-quantified arm postures, and analyzed the relationship among them, to guide students to improve their performance.

Ø  The paper is well written.

Ø  The text is clear and easy to read.

Ø  The conclusions of the manuscript are in accordance with the evidence as well as the arguments presented by the authors.

Ø  The authors address the central question quite well.

Based on the above:

Overall Recommendation: Accept after minor revision.

Comments and Suggestions for Authors:

·         Lines 136-137: The size of the font used in the equations is particularly small, and there is difficulty in reading them. Please increase the font size.

·         Figure 3, Figure 4, and Figure 4 are small in size and there is difficulty in reading their content. Please increase their size.

Round 2

Reviewer 2 Report

Comments and Suggestions for Authors

Objective conclusions cannot be drawn with only four subjects. The number of subjects should be increased to an equal number of experienced and inexperienced subjects, and objective verification should be conducted.

Author Response

Thank you for pointing this out. We agree with this comment. Our sample size is indeed insufficient, for both experienced and inexperienced students. We also mentioned the shortcomings in the article. In future studies, we will expand the sample size and compare the data obtained by the motion capture device with the current commonly used CPR simulators for consistency, such as depth, frequency, etc. Few researchers have focused on the assessment of arm angles, and a few studies have basically used electromyography or two-dimensional video images to assess whether the arm is straight or not. However, the arms are straight and perpendicular to the ground to ensure the quality of CPR is very important. Our research uses 3D data to more accurately evaluate specific angle values, whether the arms are straight or perpendicular to the ground. Our experiment initially explored the feasibility of using 3D data to evaluate the arm angles (straight and perpendicular to the ground), and proposed an expectation of the arm angle relationship and compression quality based on the existing data, hoping to provide a reference for other researchers and guide the improvement of subsequent research.Thanks again for your comments.